# Uncertainty-aware Graph Structure Learning

## Abstract

*Graph Neural Networks* (GNNs) have become a prominent approach for learning from graph-structured data. However, their effectiveness can be significantly compromised when the graph structure is sub-optimal. To address this issue, *Graph Structure Learning* (GSL) has emerged as a promising technique that refines node connections adaptively. Nevertheless, we identify two key limitations in existing GSL methods: 1) Most methods primarily focus on node similarity to construct relationships, while overlooking the quality of node information. Blindly connecting low-quality nodes and aggregating their ambitious information can degrade the performance of other nodes. 2) The constructed graph structures are often constrained to be symmetric, which may limit the model's flexibility and effectiveness.

To overcome these limitations, we propose an **Uncertainty-aware Graph Structure Learning** (UnGSL) strategy. UnGSL estimates the uncertainty of node information and utilizes it to adjust the strength of directional connections, where the influence of nodes with high uncertainty is adaptively reduced. Importantly, UnGSL serves as a plug-in module that can be seamlessly integrated into existing GSL methods with minimal additional computational cost. In our experiments, we implement UnGSL into six representative GSL methods, demonstrating consistent performance improvements.

## CCS Concepts

• **Computing methodologies → Neural networks**.

## Keywords

Graph structure learning, Graph neural network, Uncertainty

**ACM Reference Format:**
Anonymous Author(s). 2018. Uncertainty-aware Graph Structure Learning. In *Proceedings of Make sure to enter the correct conference title from your rights confirmation emai (Conference acronym 'XX)*. ACM, New York, NY, USA, 12 pages. https://doi.org/XXXXXXX.XXXXXXX

## 1 INTRODUCTION

Graph neural networks (GNNs) [1–3] have demonstrated remarkable performance in tackling graph-structured data. To date, GNNs have evolved with increasingly sophisticated model architectures [4, 5] to enhance their capabilities. However, these model-centric methods often neglect potential flaws in the underlying graph structure, which can lead to suboptimal performance. In practice, graph

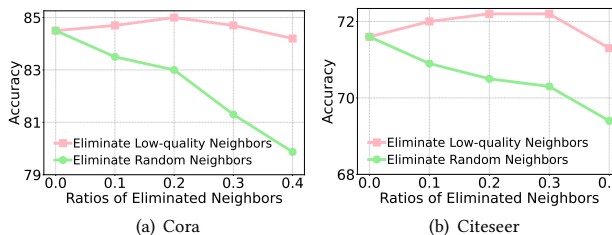

(a) Cora      (b) Citeseer

**Figure 1: Performance varies with the ratios of eliminated neighbors on Cora and Citeseer datasets. Here we first constructed a graph based on GRCN. We then eliminate a certain ratio of neighbors with the highest uncertainty for each node, and evaluate the performance of GCN on such a pruned graph. For comparison, we also report the performance under random elimination.**

data frequently exhibit suboptimal characteristics, such as noisy connections and incomplete information, due to the inherent complexities and inconsistencies in data collection [6, 7].

To address these issues, Graph Structure Learning (GSL) [8–11], a data-centric approach, has garnered increasing attention. Beyond learning node representations with GNNs, GSL learns to refine node connections and edge weights. This approach has been shown to effectively enhance the accuracy of GNNs on downstream tasks while improving their resilience to topological perturbations [6, 12]. Early work on GSL directly treated the graph structure (*i.e.,* the adjacency matrix) as learnable parameters. However, due to the large parameter space, these strategies often incur substantial computational overhead and are difficult to train effectively [8, 13]. More recently, research has shifted towards embedding-based GSL [9, 14–16], which constructs the adjacency matrix based on the similarity of node embeddings. Various similarity metrics, such as cosine similarity [15, 16] or neural networks [14], have been employed. These methods aim to increase graph homophily and typically achieve state-of-the-art performance, as nodes with similar features (or embeddings) are more likely to be connected.

Despite their success, we identify two key limitations in these embedding-based GSL methods:

• **These methods mainly rely on embedding similarity for graph construction while neglecting the quality of node information.** Given the critical role of edges in GNNs as conduits for information propagation, it is essential to evaluate the quality of the information being propagated. Aggregating unclear or ambiguous information from neighbor nodes can disrupt the embedding learning of the target node. Constructing connections based solely on node similarity, without assessing the quality of the node's information, may lead to suboptimal performance. To empirically validate this point, we conducted a simple experiment using a representative GSL method (GRCN [17]). As shown in Fig. 1, removing a certain proportion of neighbors with the highest uncertainty results in a significant performance gain.

**Figure 2: Illustration of how our UnGSL differs from existing embedding-based GSL methods. Existing GSL learns symmetric relationships, leading to a dilemma when managing connections between high-quality and low-quality nodes. In contrast, UnGSL learns asymmetric relationships, allowing low-quality nodes to benefit from high-quality nodes while mitigating the negative influence of low-quality nodes.**

- **These methods often generate symmetric graph structures, which can hinder their effectiveness.** Current embedding-based methods tend to construct symmetric relationships between nodes, implying that both nodes exert an equal and bidirectional influence during the GNN learning process. This imposed symmetry can constrain the model's flexibility and capacity, particularly when the connected nodes differ in quality. For example, consider a scenario where a high-quality node is linked to a low-quality node (refer to Fig. 2). While the high-quality node can provide valuable information that greatly benefits the low-quality node, the reverse influence from the low-quality node may have a negative impact on the high-quality node. Symmetric relationships fail to account for this disparity, leading to a dilemma in the learning process. This inspires us to explore asymmetric structure learning. By modeling directional relationships separately, we allow the low-quality node to benefit from the high-quality node's information while reducing the adverse influence in the opposite direction, thus protecting the high-quality node from negative effects. Although a few studies [11, 18, 19] have begun to explore asymmetric graph structure learning, they typically restrict the asymmetry to relations between labeled and unlabeled nodes, overlooking the richer relationships between the vast majority of unlabeled nodes.

To overcome these limitations, we propose an uncertainty-aware graph structure learning (UnGSL) method that considers nodes' information quality to learn an asymmetric graph structure. UnGSL directly utilizes the uncertainty (Shannon Entropy [20] ) of the node in classification to indicate the node's information quality, and conducts theoretical analyses to demonstrate that aggregating neighbors with higher uncertainty would increase the target node's own uncertainty. Building on this, UnGSL leverages a learnable node-wise threshold to differentiate low-quality neighbors from high-quality ones, and adaptively reduces directional edge weights from those low-quality neighbors. Notably, our UnGSL is simple and can be easily incorporated into various embedding-based GSL methods, boosting their performance with minor extra computational overhead.

In summary, this work makes the following contributions:

- We highlight the neccesity of modeling node's uncertainty in graph structure learning, and theoretically demonstrate that the uncertainty of a node after GNN layer is positively correlated with those of its neighbors.
- We propose a simple yet novel uncertainty-aware graph structure learning strategy (UnGSL), which can be seamlessly integrated with various embedding-based GSL models to mitigate the directional impact of high-uncertainty nodes.
- We conduct extensive experiments to demonstrate that UnGSL can consistently boost existing embedding-based GSL models across five benchmark datasets, with an average performance increase of 2.18%.

## 2 PRELIMINARIES

In this section, we introduce basic notations and backgraound on GNNs and GSL methods.

Consider the graph $\mathcal{G} = (\mathcal{V}, \mathcal{E}, \mathbf{A}, \mathbf{X})$, where $\mathcal{V}$ represents a set of $n$ nodes $\{v_1, ..., v_n\}$ and $\mathcal{E}$ represents the set of edges. Let $\mathbf{A}$ be the initial adjacency matrix of the graph, where $\mathbf{A}_{ij} = 1$ if an edge exists between node $v_i$ and $v_j$; otherwise, $\mathbf{A}_{ij} = 0$. The matrix $\mathbf{X} = [x_1, ..., x_n] \in \mathbb{R}^{n \times d}$ represents the node feature matrix, where each column $x_i$ corresponds to the feature vector of node $v_i$. Let $\mathbf{D}$ denote the diagonal degree matrix defined as $\mathbf{D}_{ii} = 1 + \sum_j \mathbf{A}_{ij}$; and $\hat{\mathbf{A}}$ denotes the normalized adjacency matrix with self-loop, *i.e.*, $\hat{\mathbf{A}} = \mathbf{D}^{-\frac{1}{2}}(\mathbf{A}+\mathbf{I})\mathbf{D}^{-\frac{1}{2}}$ for symmetric normalization or $\hat{\mathbf{A}} = \mathbf{D}^{-1}(\mathbf{A}+\mathbf{I})$ for row normalization.

### 2.1 Graph Neural Networks

Graph Neural Networks (GNNs) have become a prominent approach for learning from graph-structured data, where node representations are learned by iteratively aggregating and transforming information from neighboring nodes. In recent years, various designs for the aggregation and transformation processes have given rise to different GNN models[1, 2, 21]. Among these, the Graph Convolutional Network (GCN) [1] stands out as one of the most widely adopted and influential architectures. The operation at the $l$-th layer in GCN can be formulated as:

$$\mathbf{Z}^{(l)} = \sigma(\hat{\mathbf{A}}\mathbf{Z}^{(l-1)}\mathbf{W}^{(l)}), \tag{1}$$

where $\sigma(\cdot)$ denotes the activation function, $\mathbf{W}^{(l)}$ is a learnable parameter matrix used to transform the node features.

Given the critical role of the graph structure in GNNs, which determines the sources of information aggregation, ensuring the quality of graph structure is of paramount importance. Recent work demonstrates that suboptimal graph structures, even with the introduction of a small percentage of noisy edges or topological perturbations (*e.g.*, 10%), can significantly degrade the performance of GNNs (*e.g.*, 25%).[6, 12].

### 2.2 Graph Structure Learning

Graph Structure Learning (GSL) aims to enhance the accuracy and robustness of GNNs by learning the optimal graph $\mathbf{S}$ and the corresponding node representations $\mathbf{Z}^*$. Given labels $\mathbf{Y}$, the loss

function in GSL methods can be formulated as:

$$\mathcal{L} = \mathcal{L}_{Task}(\mathbf{Z}^*, \mathbf{Y}) + \lambda \mathcal{L}_{Reg}(\mathbf{Z}^*, \mathbf{S}, \mathcal{G}), \tag{2}$$

where $\mathcal{L}_{Task}$ optimizes the GNN encoder for the downstream task and $\mathcal{L}_{Reg}$ regularizes the learned adjacency matrix $\mathbf{S}$. $\lambda$ is a trade-off hyperparameter.

Traditional GSL methods [8, 13] that treat each edge $\mathbf{S}_{ij}$ as a learnable parameter often suffer from significant computational overhead and are challenging to train efficiently. Recent research has focused on embedding-based GSL methods [9, 14–16], which construct the adjacency matrix $\mathbf{S}$ by leveraging the similarity between node embeddings:

$$\mathbf{S}_{ij} = \mathbf{S}_{ji} = \phi(\mathbf{Z}_i, \mathbf{Z}_j). \tag{3}$$

Here $\phi(\cdot)$ is a metric function used to calculate the similarity between nodes.

Although the structure modeling paradigm in Eq. 3 is widely employed in GSL methods, it suffers from two key limitations:

- **This paradigm relies on embedding similarity while neglecting the quality of node information.** Only semantic similarities between embeddings $\mathbf{Z}_i$ and $\mathbf{Z}_j$ are considered in this structure modeling paradigm, while the varying uncertainties of them, which reflect their information quality, are neglected. This may undermine the quality of embeddings of target nodes when aggregating inferior information from low-quality neighbors (as validated by the preliminary experiment in the Introduction).
- **This paradigm constraining the graph to be symmetric, which potentially hinder effectiveness of GSL models.** The pair-wise similarity constrains the learned edge between $v_i$ and $v_j$ to be bidirectional, overlooking their unequal influence due to varying information quality. For example, if $\mathbf{Z}_i$ contains higher-quality information than $\mathbf{Z}_j$, the constructed edge $\mathbf{S}_{ji}$ can provide valuable information that greatly benefits the low-quality node $v_j$ while the edge $\mathbf{S}_{ij}$ propagate inferior information to poison the embdding of $v_i$. By modeling directional relationships separately, we enable the low-quality node to benefit from the high-quality node's information while mitigating the negative impact in the reverse direction.

Although a few works [11, 18, 19] have been proposed to learn asymmetric graphs by generating directed edges $\mathbf{S}_{ij}$ from the labeled node $v_j$ to unlabeled node $v_i$ and constraining $\mathbf{S}_{ji} = 0$, which facilitates the propagation of label information and avoids introducing inconsistency to the labeled nodes. However, these methods completely rely on annotated labels and fail to learn reasonable asymmetric connections between the vast majority of unlabeled nodes.

Given the flaws of existing methods, we argue for the necessity of incorporating node uncertainty into graph structure learning to learn an optimal asymmetric structure. We propose the uncertainty-aware graph structure learning (UnGSL) method to enhance GSL models, which leverage learnable node-wise thresholds to identify high-uncertainty neighbors and adaptively reduce directional edge weights from those low-quality neighbors.

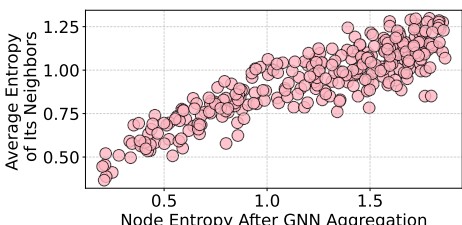

**Figure 3: Visualization of node entropy after GNN aggregation (i.e., $H(p_i)$) alongside the average entropy of its neighbors (i.e., $\sum_{v_j \in \mathcal{N}(v_i)} \hat{\mathbf{A}}_{ij} H(p'_j)$) on Cora datasets.**

## 3 METHODOLOGY

In this section, we first conduct theoretical and empirical analyses to demonstrate the detrimental impact of neighbors with high uncertainty levels on GNN learning (Subsection 3.1). We then present the proposed Ucertainty-aware graph structure learning method in detail (Subsection 3.2).

### 3.1 Analyses on the Impact of Neighbor Uncertainty

Aggregating unclear or ambiguous information can intuitively disrupt the learning process of target nodes, thereby negatively affecting the performance of Graph Neural Networks (GNNs). In this section, we aim to conduct both theoretical and empirical analyses to substantiate this claim. To begin, we introduce several formal concepts to facilitate these analyses.

**Semi-supervised Node Classification Task.** For convenience, we refer to recent analytical work on GNNs [22] and focus our theoretical analysis on the semi-supervised node classification task, which is the most common and widely studied scenario. Nevertheless, at the end of this section, we will also discuss how our method can be adapted to the unsupervised learning scenario. Following the definitions in [22], we consider a $K$-class classification problem and employ a linear classification model. Formally, the classification logits can be expressed as:

$$\mathbf{O} = \mathbf{D}^{-1}(\mathbf{A} + \mathbf{I})\mathbf{X}\mathbf{W}, \tag{4}$$

where $\mathbf{W} \in \mathbb{R}^{d \times K}$ is the linear classification matrix. Assume the logit in matrix $\mathbf{O}$ are bounded by the scalar 1, *i.e.,* $\max |\mathbf{O}_{ij}| < 1$. From a local perspective for node $v_i$, its probabilities can be formulated as :

$$p_i = \frac{\mathbf{O}_i + \mathbf{1}_K}{\sum_{j=1}^K (\mathbf{O}_{ij} + 1)}, \tag{5}$$

where $\mathbf{1}_K$ is $K$-dimensional all-ones vector. Here we simple omit the nonlinear exponential function in the softmax, given that our primary focus is on the impact of aggregation on node uncertainty and the exponential function mainly serves to generate positive values. This simplification has been adopted in prior work [23, 24], and it also can be considered a first-order Taylor approximation.

**Uncertainty Estimation via Entropy.** Entropy measures the information uncertainty within a probability distribution [20] . For node classification tasks, the entropy of the classification probabilities reflects the classifier's certainty in assigning the node representation to a specific class. A high-entropy node indicates that

its representation carries significant uncertainty, making it challenging for the classifier to reach a confident decision. Aggregating information from such nodes can poison the target node's representation, hindering the generation of accurate predictions. Given probabilities $p_i$ of node $v_i$, its entropy is defined as:

$$H(p_i) = -\sum_{k=1}^{K} p_{ik} \log(p_{ik}). \tag{6}$$

We further discuss the uncertianty metric for unsupervised learning scenario at the end of this section.

Formally, to demonstrate the impact of the uncertainty of neighbors along GNNs, we have the following proposition, with detailed proof provided in Appendix A:

PROPOSITION 1. *Define the logits of the initial node feature matrix as* $\mathbf{O}' = \mathbf{XW}$. *For a given node* $v_i$, *let* $p_i$ *denote its classification probabilities after GNN aggregation.* $\forall v_j \in \mathcal{N}(v_i)$, *let* $p'_j$ *denote classification probabilities of its initial features. Then the entropy of* $v_i$ *and the entropy of* $v_j \in \mathcal{N}(v_i)$ *satisfy the following inequality:*

$$H(p_i) \geq \sum_{v_j \in \mathcal{N}(v_i)} \eta_j H(p'_j), \tag{7}$$

*where*

$$\eta_j = \frac{\sum_{k=1}^{K} \hat{\mathbf{A}}_{ij}(O'_{jk} + 1)}{\sum_{v_j \in \mathcal{N}(v_i)} \sum_{k=1}^{K} \hat{\mathbf{A}}_{ij}(O'_{jk} + 1)}. \tag{8}$$

**Discussion.** Accoding to Proposition 1, the entropy of a node is lower-bounded by the weighted sum of its neighbors' entropies prior to aggregation. We can therefore obtain the following insight: If a node is connected to high-uncertainty neighbors, its own uncertainty will inevitably increase after GNN learning, degrading its representation and leading to incorrect node classification.

**Empirical Analyses.** We conduct a simple experiment to empirically demonstrate that aggregating neighbors with higher uncertainty would increase the node's own uncertainty. Specifically, we train the model with 1-layer GCN and linear classifier on given datasets. Then we visualize the entropy of a node after GNN aggregation (i.e., $H(p_i)$) alongside the average entropy of its neighbors (i.e., $\sum_{v_j \in \mathcal{N}(v_i)} \hat{\mathbf{A}}_{ij} H(p'_j)$). As shown in Fig. 3, we observe a strong linear correlation between the entropy of a node after GNN aggregation and the average entropy of its neighbors, which substantiates our proposition( see additional experimental results in the Appendix C.1) .

The above analyses clearly illustrates the impact of node uncertainty in aggregation process in GNNs. Blindly connecting and aggregating information from nodes with high uncertainty may undermine the performance of the nodes themselves. It is therefore important to consider node uncertainty in graph structure learning to learn a reasonable asymmetric structure. Specifically, one can prevent a node falling into a high-uncertainty region by weakening its connections to neighbors with high uncertainty. Meanwhile, it can receive more stable information via strengthened connections with low-uncertainty nodes.

## 3.2 Uncertainty-aware Graph Structure Learning

Given the importance of considering uncertainty in graph structure learning, we propose the simple yet novel uncertainty-aware graph structure learning (UnGSL) method that leverages learnable node-wise thresholds to distinguish low-quality neighbors from high-quality ones and adaptively refines edges based on their uncertainty levels. Specifically, UnGSL first pretrains the GSL model to estimate node uncertainty. Afterwards, it normalizes these uncertainty into confidence scores, which are used to construct a node confidence matrix. Then, UnGSL applies learnable node-wise thresholds to split neighbors with different levels of uncertainty into two groups (i.e., high-confidence and low confidence) for each node. Finally, it amplifies edge weights from confident neighbors while reducing edge weights from uncertain ones. In summary, the process of UnGSL can be formulated as:

$$\mathbf{C} = e^{-\mathbf{U}} \cdot \mathbf{1}_n^\top, \tag{9}$$

and

$$\mathbf{A}^* = \mathbf{S} \odot \psi(\mathbf{C} - \boldsymbol{\varepsilon} \cdot \mathbf{1}_n^\top), \tag{10}$$

where $\mathbf{U} = [u_1, ..., u_n]$ is node uncertianty estimated during pretraining stage, $\boldsymbol{\varepsilon} = [\varepsilon_1, ..., \varepsilon_n]$ is learnable node-wise thresholds, $\mathbf{C}$ is node confidence matrix, $\psi(\cdot)$ is a activate function, $\mathbf{S}$ is the construct graph of the GSL model, and $\odot$ denotes the Hadamard product operator.

Here Eq. 9 aims to normalize node uncertainty into confidence within the interval $(0, 1)$ and transform them into a node confidence matrix $\mathbf{C}$, where $\mathbf{C}_{ij}$ is the confidence of node $v_j$.

Eq. 10 leverage node-wise learnable thresholds $\boldsymbol{\varepsilon}$ to distinguish low-confidence neighbors (i.e., $\mathbf{C}_{ij} < \varepsilon_i$) from high-confidence ones (i.e., $\mathbf{C}_{ij} \geq \varepsilon_i$) and adaptively refines the corresponding edges using the following activation function:

$$\psi(x) = \begin{cases} \tau \cdot s(x), & x \geq 0, \\ \beta, & x < 0, \end{cases} \tag{11}$$

where $s(\cdot)$ is the sigmoid function, $\tau$ is a hyperparameter amplifies of edge weights from high-confidence neighbors, $\beta$ is a hyperparameter that controls the reduction of edge weights from low-confidence neighbors.

Based on the above formulation, we next discuss several key advantages of UnGSL:

**Uncertainty-aware.** UnGSL considers the information quality of nodes during the structure modeling process, facilitating the learning of an optimal graph that mitigates the negative impact of high-uncertainty neighbors in GNN learning.

**Asymmetric Graph Structure.** UnGSL proposes learning an asymmetric graph, where the edge weights between nodes differ based on their uncertainty levels. Specifically, UnGSL weakens edges from uncertain nodes to confident nodes, mitigating the impact of inferior information. Meanwhile it strengthens the edge in the opposite direction to improve the representations of uncertain nodes.

Notably, several methods that construct directed edges from labeled nodes to unlabeled nodes [11, 19] can be viewed as specific cases of UnGSL, as the embeddings of labeled nodes are directly optimized during training and therefore more likely to exhibit lower

uncertainty. In contrast, UnGSL models the asymmetric relationships among the predominantly unlabeled nodes, leading to superior performance. We further empirically validate this in Section 4.

**Model-agnostic.** UnGSL can be seamlessly integrated with various GSL models to further enhance their ability to learn graphs, improving the performance of GNNs on downstream tasks. Comprehensive experiments supporting this can be found in Section 4.

**Efficiency.** UnGSL contains only $n$ learnable parameters, imposing minimal burden on the training of GSL models. Besides, UnGSL refines the existing edges of the given graph without generating new ones, resulting in slight impact on the space consumption of constructed adjacency matrix.

**Adaptive to Unsupervised Scenarios.** Despite the mainstream focus of GSL research on supervised learning, our approach can be generalized to unsupervised GSL scenarios. The main challenge lies in determining an appropriate uncertainty metric for unsupervised GSL method. We suggest employing the self-supervised structure learning loss (*i.e.,* graph contrastive learning loss) as uncertainty. The GCL loss can measure the invariance of node representations to feature or structure perturbations, where this invariance can be interpreted as the uncertainty of nodes with respect to their original features and structure.

## 4 EXPERIMENTS

In this section, we conduct experiments to evaluate the effectiveness of the proposed UnGSL strategy. Our experiments aim to answer the following research questions:

- **RQ1:** Does the integration of UnGSL into existing GSL methods lead to performance improvements?
- **RQ2:** What is the impact of key configurations (e.g., node-wise thresholds, asymmetric graphs, hyperparameters $\beta$) on UnGSL performance?
- **RQ3:** Can UnGSL enhance the robustness of GNNs against structural perturbations?
- **RQ4:** How well does UnGSL generalize across different GNN backbones?
- **RQ5:** Does UnGSL introduce significant additional computational overhead?

### 4.1 Experiment Settings

*4.1.1 Datasets.* To comprehensively evaluate UnGSL's performance on node classification, we follow previous works [6, 17] and select 5 commonly used datasets, including three homophilous citation datasets [25] (Cora, Citeseer, and Pubmed) and two heterophilous datasets (Blogcatalog [26] and Roman-Empire [27]). The chosen datasets cover a wide range of homophily levels and graph sizes, allowing us to demonstrate UnGSL's effectiveness under various conditions. For a fair comparison, we strictly follow the data split settings used in the newly proposed benchmark for GSL [6]. Detailed statistics of these datasets are provided in Appendix B.

*4.1.2 Baselines.* To demonstrate UnGSL's generalizability across different GSL models, we select 6 state-of-the-art GSL algorithms corresponding to the chosen datasets as baselines:

- GRCN [17] uses a GCN to extract topological features and compute the similarity between nodes as the edge weights of the learned graph.
- ProGNN [8] treats the graph as a learnable adjacency matrix and optimizes the sparsity, low-rankness, and feature smoothness of the graph structure.
- PROSE [15] identifies influential nodes using PageRank scores and reconstructs the graph structure by connecting these influential nodes.
- IDGL [9] iteratively learns the graph structure and node embeddings, and introduces a node-anchor message-passing paradigm to scale IDGL to large graphs.
- SLAPS [10] proposes learning a denoising autoencoder to filter noisy edges in the graph structure.
- CUR [11] is a model-agnostic structure learning module, which proposes constructing unidirectional edges from unlabeled nodes to labeled nodes via CUR decomposition to facilitate the propagation of supervision signals to unlabeled nodes.
- SUBLIME [28] is an unsupervised GSL model that employs contrastive learning between the learned graph and an augmented graph to enhance the robustness of the graph structure.

For all models, we report the average performance and standard deviations of 10 runs with different random seeds.

*4.1.3 Configuration.* With regard to hyperparameter setting, we tuned the hyperparameters of UnGSL using Bayesian search. Specifically, the initial values of the learnable nodewise thresholds $\varepsilon$ are uniformly set within the range (0, 1). The hyperparameter $\beta$ was searched within the range (0, 1). For hyperparameter $\tau$, we found that setting it to a constant value of 2 works well. For detailed hyperparameter settings please refer to Appendix C.4.

We optimized the model using Adam optimizer, with the learning rate selected from range (0.01, 0.0001). For all baselines, we strictly adhere to their original settings for hyperparameter tuning to ensure that they attain best performance. All GSL methods are evaluated based on the performance of GNNs on downstream tasks when using the learned structure. We also consider cross-architecture scenarios in Section 4.5, where GSL training and downstream tasks use different GNN architectures.

### 4.2 Main Results (RQ1)

*4.2.1 Comparison to Vanilla GSL Models.* Table 1 presents the experimental results of the UnGSL module applied to various GSL models. We can observe that: 1) UnGSL significantly improves the node classification accuracy for all GSL models across all datasets with an average increase of 2.18%, achieving new state-of-the-art performance in the GSL literature. The improvement brought by UnGSL is particularly pronounced on GRCN, achieving an average improvement of 4.70%. These results empirically demonstrate UnGSL's effectiveness in further denoising the learned graph structure from the perspective of uncertainty, resulting in more reliable node representations and accurate predictions. 2) For the self-supervised GSL model SUBLIME, UnGSL continues to achieve higher accuracy by utilizing contrastive loss as a form of uncertainty estimation to guide structure refinement, resulting in an average improvement of 2.35%.

**Table 1: Node classification accuracy±std comparsion(%). Each experiment was repeated 10 times with different random seeds. "OOM" denotes out of memory. The top-performing results are marked in bold.**

| Model | Cora | Citeseer | Pubmed | BlogCatalog | Roman-empire |
|-------|------|----------|--------|-------------|--------------|
| GRCN | 84.70±0.31 | 72.49±0.77 | 78.94±0.16 | 76.17±0.23 | 44.29±0.28 |
| GRCN+UnGSL | **85.84±0.51** | **73.88±0.55** | **79.59±0.48** | **76.78±0.11** | **52.54±0.31** |
| PROGNN | 80.39±0.41 | 67.94±0.52 | OOM | 76.17±0.22 | OOM |
| PROGNN+UnGSL | **81.86±0.55** | **69.66±0.27** | OOM | **76.82±0.19** | OOM |
| PROSE | 81.1±0.45 | 72.3±0.37 | 83.3±0.71 | 75.31±0.17 | 55.61±0.34 |
| PROSE+UnGSL | **81.90±0.31** | **73.10 ±0.32** | **83.86 ±0.30** | **75.77±0.36** | **56.17±0.31** |
| IDGL | 84.50±0.5 | 72.49±0.67 | 82.83±0.33 | 89.66±0.28 | 46.67±0.56 |
| IDGL+UnGSL | **84.90±0.42** | **73.74±1.01** | **83.33±0.32** | **92.13±0.18** | **47.05±0.59** |
| SLAPS | 72.89±1.02 | 70.05±0.83 | 69.12±1.00 | 91.62±0.39 | 63.42±0.24 |
| SLAPS+UnGSL | **74.28±0.95** | **72.08±0.94** | **70.32±1.04** | **91.89±0.41** | **64.33±0.29** |
| SUBLIME | 82.50±0.6 | 71.56±0.17 | 80.41±0.69 | 93.39±0.24 | 63.48±0.53 |
| SUBLIME+UnGSL | **84.24±0.91** | **74.34±0.73** | **80.84±0.92** | **95.37±0.22** | **65.45±0.32** |

**Table 2: Performance comparison between the CUR decomposition and UnGSL modules.**

| Model | Cora | Citeseer | Roman-empire |
|-------|------|----------|--------------|
| GRCN+CUR | 84.89±0.22 | 73.35±0.46 | 44.04±0.28 |
| GRCN+UnGSL | **85.47±0.40** | **73.93±0.45** | **52.46±0.32** |
| IDGL+CUR | 84.73±0.23 | 72.93±0.85 | OOM |
| IDGL+UnGSL | **84.89±0.22** | **74.73±0.60** | **46.68±0.63** |

**Table 3: Ablation study on the UnGSL when integrating with GRCN model. We report the performance of UnGSL and two variants.**

| Method | Cora | Citeseer | Roman-empire |
|--------|------|----------|--------------|
| Fixed $\varepsilon$ | 85.23±0.15 | 73.2±0.96 | 44.72±0.14 |
| Symmetrize $A^*$ | 85.03±0.40 | 73.74±0.49 | 52.28±0.025 |
| UnGSL | **85.47±0.40** | **73.93±0.45** | **52.46±0.32** |

*4.2.2 Comparison to the Label-oriented Directed GSL Module.* Here We compare the performance of UnGSL with CUR decomposition [11], another GSL module proposed recently to learn asymmetric graph structure by constructing directed edges from labeled nodes to unlabeled nodes. Table 2 shows the experiment results of UnGSL and CUR decomposition. We can observe that: 1) UnGSL consistently outperforms CUR decomposition on supervised GSL methods. 2) UnGSL demonstrates better scalability with large-scale graphs compared to CUR decomposition. For example, IDGL+UnGSL is applicable to the Roman-Empire dataset and further enhances accuracy, while IDGL+CUR encounters an out-of-memory problem.

## 4.3 Ablation Studies (RQ2)

*4.3.1 Effects of Adaptive Threshold $\varepsilon$.* To highlight the effectiveness of the learnable threshold, we considered a variant where we fixed the $\varepsilon$ in UnGSL during training phase. Specifically, for each node, we select a fixed proportion of the most uncertain neighbors, as high-uncertainty neighbors, where this proportion is consistent across all nodes. Next, we reweight edges according to Eq. 9. As shown in

Table 3, the learnable $\varepsilon$ outperforms the fixed $\varepsilon$ across 3 datasets. The results indicate that the learnable threshold effectively differentiates high-uncertainty neighbors from low-uncertainty neighbors for each node. By adaptively modifying the corresponding edges, UnGSL further enhances node representations and facilitates accurate predictions.

*4.3.2 Superiority of Asymmetric Graph.* To demonstrate the superiority of the asymmetric edges constructed by UnGSL, we symmetrize the graph refined by UnGSL during training phase, generating an symmetric graph structure. As shown in Table 3, symmetrizing graph $A^*$ of UnGSL degrades its performance on node classification. The underlying reason is that the symmetrization operation may perturb the graph by weakening edges from low-uncertainty neighbors and strengthening edge weights from high-uncertainty neighbors, which violates the core mechanism of UnGSL.

*4.3.3 Analysis on Hyperparameter $\beta$.* To explore the role of $\beta$ in UnGSL's activation function $\psi(\cdot)$, we assign different values of $\beta$ from interval $[0, 1]$ and evaluate the corresponding performance across various GSL models. As shown in Figure 4, we can observe that: (1) UnGSL enhances the accuracy of GSL models by reducing edge weights to low-uncertainty neighbors using an appropriate $\beta$. (2) UnGSL with positive $\beta$ consistently outperform UnGSL with $\beta = 0$. The results indicate that high-uncertainty neighbors still contain valuable information that enhances the node's representation, and removing connections from high-uncertainty nodes blindly may lead to information loss. Furthermore, setting $\beta = 0$ in UnGSL directly reduces the average node degree, leading to increased neighbor distribution variance and consequently undermining intra-class node separability [22].

## 4.4 Robustness Analysis (RQ3)

To assess the robustness of UnGSL in GSL models under topological perturbations, we randomly added or removed edges from the original Cora and Blogcatalog datasets. The ratio of modified edges varied from 0 to 0.8 to simulate different levels of perturbation

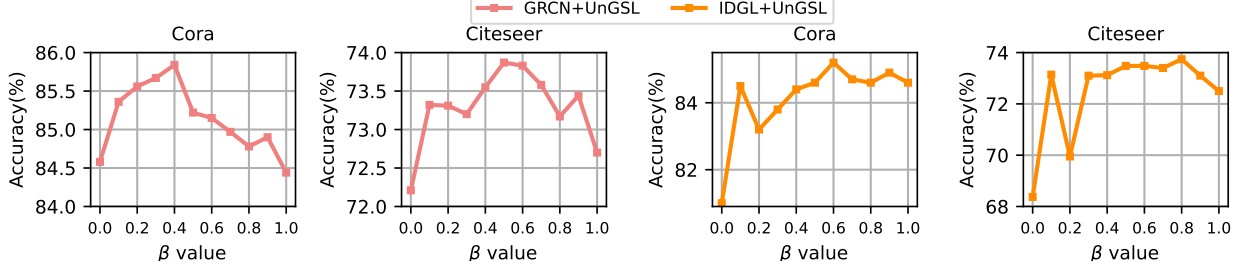

Figure 4: Comparsion of different $\beta$ on Cora and Citeseer datasets.

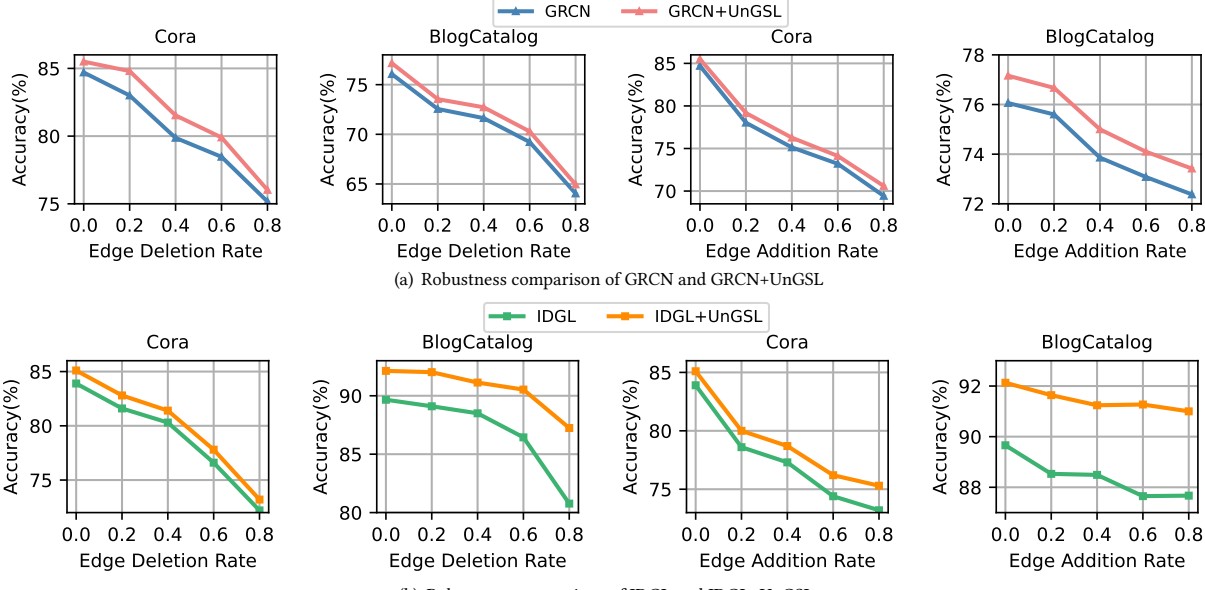

(a) Robustness comparison of GRCN and GRCN+UnGSL

(b) Robustness comparison of IDGL and IDGL+UnGSL

Figure 5: Robustness analysis with random noise injection on Cora and BlogCatalog datasets.

intensity. As shown in Figure 5, all GSL models experience a performance decline as the ratio of modified edges increases, while UnGSL can generally improves the performance of GSL models across different perturbation ratios, demonstrating its robustness against topological perturbations.

## 4.5 Generalizability on GNN Models (RQ4)

We further consider the scenario where GSL training and downstream tasks use different GNN backbones. We evaluate the generalizability of the learned structures generated by the GSL+UnGSL on several other GNN models, including SGC [2] , APPNP [29], GAT[21], and JKNet [30]. The results are presented in Table 4. We observe that the graphs produced by GSL+UnGSL improve the prediction of various GNN models compared to those generated by vanilla GSL. Overall, UnGSL demonstrates its generalizability in enhancing performance across different GNN architectures.

## 5 Effiency Analysis (RQ5)

We analysis the time and memory effiency of UnGSL on Cora dataset. To assess time efficiency, we evaluate the algorithms by

measuring the time taken to converge, *i.e.,* to reach optimal performance on the validation set. As shown in Fig. 6, while GSL models with UnGSL slightly increase convergence time and training space, it can reduce SUBLIME's convergence time. In general, the results show that UnGSL slightly increases the time and space costs of GSL models, demonstrating its efficiency (see additional experimental results in the Appendix C.2) .

## 6 RELATED WORK

In this section, we provide a brief review of related work from three perspectives.

### 6.1 Graph Neural Networks

Graph neural networks (GNNs) are powerful models for learning node representations from graph data. Existing GNNs can be categorized as spectral GNNs and spatial GNNs. Spectral GNNs leverage eigenvectors and eigenvalues within the graph Laplacian matrix to design graph signal filters in the spectral domain [31–34]. Spatial GNNs [1–3, 21, 35] simplified spectral graph filter [32] using

**Table 4: Generalizability of UnGSL with different backbones on Cora and Citeseer datasets. The value in bold signifies the top-performing result.**

| Methods | Cora | | | | Citeseer | | | |
|---|---|---|---|---|---|---|---|---|
| | SGC | APPNP | GAT | JKNet | SGC | APPNP | GAT | JKNet |
| GRCN | 84.40±0.00 | 84.00±0.15 | 81.45±1.04 | 83.73±1.33 | 72.00±0.00 | 72.73±0.96 | 70.77±1.06 | 71.83±0.72 |
| GRCN+UnGSL | **84.80±0.10** | **84.40±0.61** | **82.45±1.04** | **84.53±1.38** | **72.37±0.06** | **73.50±0.30** | **70.88±1.70** | **72.32±1.75** |
| IDGL | 78.00±0.00 | 82.13±0.32 | 79.87±1.01 | 76.23±1.19 | 71.90±0.00 | 72.20±0.75 | 63.83±0.97 | 71.00±1.15 |
| IDGL+UnGSL | **78.20±0.12** | **82.50±0.61** | **80.23±0.49** | **77.10±0.50** | **73.40±0.00** | **73.37±0.61** | **67.00±0.69** | **72.33±0.9** |

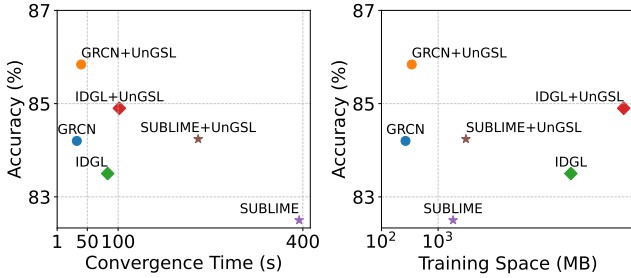

**Figure 6: Time and space consumption of different methods on Cora dataset.**

first-order approximation, which aggregates features from neighboring nodes in the spatial graph to generate node embeddings. In recent years, GNNs have developed sophisticated architectures to handle complex task scenarios, such as structural distribution shifts [36], continual graph learning [37], and model explainability [38]. However, existing GNNs assume that the input graph structure is sufficiently clean for learning, whereas real-world graphs are often noisy and incomplete, which limits GNN performance on downstream tasks [39]. In this paper, we propose uncertainty-aware graph structure learning, which effectively denoises the graph structure to alleviate the above limitation.

### 6.2 Graph Structure Learning

Graph Structure Learning (GSL) aims to learn an optimal graph that improves the accuracy of Graph Neural Networks (GNNs) on downstream tasks while enhancing their robustness against topological perturbations. Early GSL methods [8, 13] directly treat the target adjacency matrix as learnable parameters, incurring substantial computational overhead and optimization challenge. Mainstream GSL methods [9, 15–17] learn edge weights based on node-pair embedding similarities, employing various metrics such as cosine similarity [15, 16], inner product [17] or neural networks [14]. These methods aim to increase graph homophily and typically achieve state-of-the-art performance, as these metrics can capture nodes with similar semantics. However, these embedding-based GSL methods suffer from two limitations: First, they construct edges solely based on embedding similarities while neglecting node uncertainty, which may introduce inferior information that poison the target node's embedding. Although some works [14, 40] incorporate predictive uncertainty in structure learning, they only use it for

cross-view structure fusion without distinguishing nodes of varying uncertainty levels within a graph. Second, embedding-based methods impose bidirectional edges between nodes, disregarding their unequal influence due to varying levels of uncertainty. Recently, several methods propose constructing directed edges from labeled to unlabeled nodes to facilitate the propagation of supervision signals [11, 18, 19]. However, these methods fail to learn reasonable asymmetric connections between the vast majority of unlabeled nodes. Different from the above works, we propose an uncertainty-aware neighbor learning (UnGSL) strategy that learns nodewise thresholds to differentiate low-uncertainty from high-uncertainty neighbors and adaptively refine the corresponding edges. Notably, UnGSL is able to directly applied to most GSL models where edge weights updated through gradient descent, and it is not compatible with a few GSL models [41–43] that do not meet this criterion.

### 6.3 Uncertainty in GNNs

GNNs inevitably present uncertainty towards their predictions, leading to unstable and erroneous prediction results [44]. In recent years, uncertainty in GNNs has been widely researched to adapt various tasks, including out-of-distribution (OOD) detection [23, 24, 45], trustworthy GNN learning [46, 47], and GNN modeling [7]. However, these uncertainty-based GNNs assume that the input graph is sufficiently clean and primarily focus on incorporating uncertainty into the model architecture. In contrast, our proposed uncertainty-aware graph structure learning aims to refine the edge based on node uncertainty, assisting in improving graph quality.

## 7 CONCLUSION

In this paper, we first conduct theoretical and empirical analyses to demonstrate the detrimental impact of neighbors with high uncertainty levels on GNN learning. Building on this, we propose the UnGSL strategy, a lightweight plug-and-play module that integrates seamlessly with state-of-the-art GSL models and boosts performance with minimal extra computational overhead. UnGSL learns nodewise thresholds to differentiate between low-uncertainty and high-uncertainty neighbors, and adaptively refines the graph based on each node's uncertainty level. Experiments demonstrate that UnGSL consistently enhances the performance and robustness of existing GSL models. In the future, we plan to explore more effective uncertainty metrics to accurately identify uncertain nodes in graph structure learning.

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

# A  Proof of Proposition 1

To prove Proposition 1, we first introduce the log-sum inequality below.

LEMMA 1 (LOG-SUM INEQUALITY). *Let $a_1, ..., a_n$ and $b_1, ..., b_n$ be nonnegative numbers. Denote the sum of all $a_i$s by $a$ and the sum of all $b_i$s by $b$. Then log-sum inequality states that*

$$\sum_{i=1}^{n} a_i \log \frac{a_i}{b_i} \geq a \log \frac{a}{b}, \tag{12}$$

*Proof.* $\forall v_i \in \mathcal{V}$, by define the logit $o_i = \mathbf{W} \sum_{v_j \in \mathcal{N}(v_i)} \hat{\mathbf{A}}_{ij} x_j$ and logit $o'_i = \mathbf{W} x_i$, where $\sum_{v_j \in \mathcal{N}(v_i)} \hat{\mathbf{A}}_{ij} = 1$, we have:

$$o_i = \sum_{v_j \in \mathcal{N}(v_i)} \hat{\mathbf{A}}_{ij} o'_j \tag{13}$$

then for predictive probability vector $p_i$ of $o_i$, we have:

$$
\begin{aligned}
p_i &= \frac{o_i + \mathbf{1}_K}{\sum_{k=1}^{K}(o_{ik} + 1)} \\
&= \frac{\sum_{v_j \in \mathcal{N}(v_i)} \hat{\mathbf{A}}_{ij}(o'_j + \mathbf{1}_K)}{\sum_{k=1}^{K}(\sum_{v_j \in \mathcal{N}(v_i)} \hat{\mathbf{A}}_{ij} o'_j + 1)} \\
&= \frac{\sum_{v_j \in \mathcal{N}(v_i)} p'_j(\hat{\mathbf{A}}_{ij} \sum_{k=1}^{K}(o'_{jk} + 1))}{\sum_{v_j \in \mathcal{N}(v_i)} \hat{\mathbf{A}}_{ij} \sum_{k=1}^{K}(o'_{jk} + 1)}, \\
&= \sum_{v_j \in \mathcal{N}(v_i)} \eta_j p'_j, \tag{14}
\end{aligned}
$$

where, $\sum_{v_j \in \mathcal{N}(v_i)} \eta_j = 1$. Now we focus the $l$-th element in the probability vector:

$$
\begin{aligned}
p_{il} &= \sum_{v_j \in \mathcal{N}(v_i)} \eta_j p'_{jl} \\
\implies p_{il} \log p_{il} &= (\sum_{v_j \in \mathcal{N}(v_i)} \eta_j p'_{jl}) \log(\sum_{v_j \in \mathcal{N}(v_i)} \eta_j p'_{jl}) \\
&= (\sum_{v_j \in \mathcal{N}(v_i)} \eta_j p'_{jl}) \log(\frac{\sum_{v_j \in \mathcal{N}(v_i)} \eta_j p'_{jl}}{\sum_{v_j \in \mathcal{N}(v_i)} \eta_j}) \\
&\leq \sum_{v_j \in \mathcal{N}(v_i)} \eta_j p'_{jl} \log p'_{jl} \\
\implies -\sum_{l=1}^{K} p_{il} \log p_{il} &\geq -\sum_{v_j \in \mathcal{N}(v_i)} \eta_j \sum_{l=1}^{K} p'_{jl} \log p'_{jl} \\
&= \sum_{v_j \in \mathcal{N}(v_i)} \eta_j H(p'_j), \tag{15}
\end{aligned}
$$

which completes the proof.

# B  Datasets

Table 5 shows the statistics of these datasets.

# C  Additional Experimental Results

## C.1  Visualization of Node Entropy and Average Neighbor Entropy

Figure 7 presents the node entropy after GNN aggregation, alongside the average entropy of its neighbors, on the Citeseer dataset.

Table 5: Detailed statistics of node classification datasets.

| Dataset | #Nodes | #Edges | #Feat. | #Avg.degree | #Homophily |
|---|---|---|---|---|---|
| Cora | 2,708 | 5,278 | 1,433 | 3.9 | 0.81 |
| Citeseer | 3,327 | 4,552 | 3,703 | 2.7 | 0.74 |
| Pubmed | 19,717 | 44,324 | 500 | 4.5 | 0.80 |
| BlogCatalog | 5,196 | 171,743 | 8,189 | 66.1 | 0.40 |
| Roman-Empire | 22,662 | 32,927 | 300 | 2.9 | 0.05 |

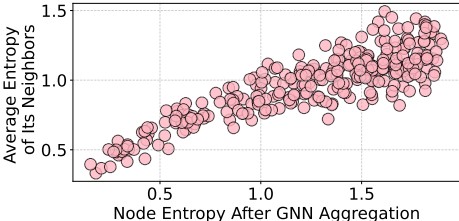

Figure 7: Visualization of node entropy after GNN aggregation (i.e., $H(p_i)$) alongside the average entropy of its neighbors (i.e., $\sum_{v_j \in \mathcal{N}(v_i)} \hat{\mathbf{A}}_{ij} H(p'_j)$) on Citeseer datasets.

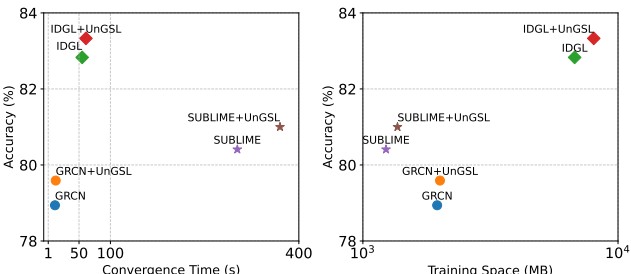

Figure 8: Effiency analysis on Pubmed dataset.

## C.2  Effiency Analysis

*C.2.1  Time and Space Effiency Analysis.* Fig. 8 shows the time and memory efficiency of UnGSL on the Pubmed dataset. the results show that UnGSL slightly increases the time and space costs of GSL models, demonstrating its efficiency.

*C.2.2  Average Degree of Learned Graph.* To demonstrate that UnGSL introduces fewer burden to the learned graph in the GSL model, we compute the average degree of learned graph before and after applying UnGSL. As shonw in Table 6, after applying UnGSL, the node degree shows a slight increase, and in some cases, the node degree even decreases compared to the original learned graph. This demonstrates the superiority of UnGSL, which uses fewer edges to significantly enhance GSL's performance.

In general, the results show that UnGSL slightly increases the time and space costs of GSL models, demonstrating its efficiency.

## C.3  Visualization on Predictive Uncertainty.

To demonstrate that UnGSL effectively reduces the predictive uncertainty of nodes in a learned graph, we analyze the changes in the entropy distribution. Specifically, we select nodes from the test sets of the Cora and Citeseer datasets, calculate their entropies,

**Table 6: Comparison of the average node degree in the learned graph structure before and after applying UnGSL.**

| Model | Cora | Citeseer | Pubmed |
|---|---|---|---|
| GRCN | 156.2 | 319.4 | 1.8 |
| GRCN+UnGSL | 156.8 | 312.9 | 2.1 |
| IDGL | 2618.2 | 3271.7 | 1390.6 |
| IDGL+UnGSL | 2649.8 | 3277.6 | 1316.6 |
| SUBLIME | 2.6 | 5.7 | 9.1 |
| SUBLIME+UnGSL | 1.1 | 5.2 | 7.4 |

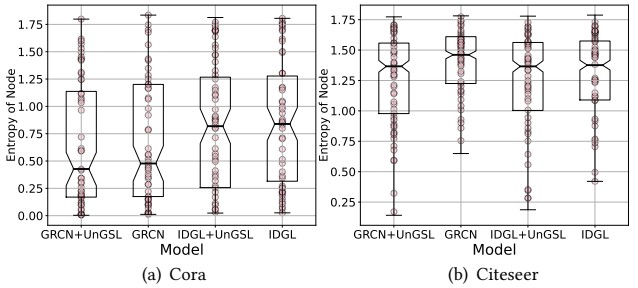

(a) Cora

(b) Citeseer

**Figure 9: Boxplots of the node entropy on Cora and Citeseer datasets.**

**Table 7: Hyperparameter settings of learning rate, $\beta$ and initial value in UnGSL.**

| | Setting | GRCN | PROGNN | PROSE | IDGL | SLAPS | SUBLIME |
|---|---|---|---|---|---|---|---|
| Cora | lr | 0.0005 | 0.0001 | 0.01 | 0.001 | 0.0001 | 0.001 |
| | $\beta$ | 0.40 | 0.82 | 0.01 | 0.95 | 0.75 | 0.98 |
| | Initial value | 0.57 | 0.48 | 0.18 | 0.71 | 0.019 | 0.20 |
| Citeseer | lr | 0.006 | 0.03 | 0.001 | 0.0005 | 0.03 | 0.03 |
| | $\beta$ | 0.56 | 0.57 | 0.75 | 0.78 | 0.19 | 0.89 |
| | Initial value | 0.47 | 0.078 | 0.66 | 0.62 | 0.41 | 0.72 |
| Pubmed | lr | 0.09 | - | 0.01 | 0.001 | 0.02 | 0.004 |
| | $\beta$ | 0.35 | - | 0.22 | 0.16 | 0.65 | 0.46 |
| | Initial value | 0.867 | - | 0.66 | 0.62 | 0.41 | 0.72 |
| BlogCatalog | lr | 0.0056 | 0.0002 | 0.06 | 0.01 | 0.007 | 0.0001 |
| | $\beta$ | 0.65 | 0.87 | 0.36 | 0.90 | 0.18 | 0.47 |
| | Initial value | 0.82 | 0.95 | 0.94 | 0.03 | 0.78 | 0.438 |
| Roman-Empire | lr | 0.0008 | - | 0.001 | 0.002 | 0.02 | 0.007 |
| | $\beta$ | 0.001 | - | 0.001 | 0.82 | 0.83 | 0.54 |
| | Initial value | 0.93 | - | 0.96 | 0.31 | 0.075 | 0.084 |

and present the results in a box plot, as shown in Fig. 9. We observe that GSL models incorporating UnGSL generally reduce node entropy, demonstrating UnGSL's effectiveness in learning high-quality structures that prevent nodes from falling into regions of high uncertainty.

## C.4 Hyperparameter Settings

Table 7 presents the hyperparameter settings of learning rate, $\beta$ and the initial value in UnGSL.

