# OpenReview forum: "Uncertainty-aware Graph Structure Learning"
_ACM.org/TheWebConf/2025/Conference — WWW 2025 Poster_

### Official Review · Reviewer_kadT · 2024-11-19

**Novelty:** 5
**Technical Quality:** 4

**Review:**

This paper provides a directed reweighting scheme for the edges in graph. Given any pretrained GSL model, UnGSL strengthen connectivity from nodes with low uncertainty and weaken connectivity from nodes with high uncertainty. The idea of utilizing uncertainty as the quality of node is novel. This paper is well written and easy to follow. The authors conduct extensive experiments to support their claim.


Pros:
- The Uncertainty-aware Graph Structure Learning is well motivated though PROPOSITION 1.
- The idea is intuitive and easy to understand.
- The experiments are extensive and thorough.
- UnGSL works in both semi-supervised and unsupervised case.

Cons:
- The meaning of ‘quality’ is unclear. Does low quality only indicates high uncertainty (entropy)?
- $\tau$ is set to be 2 as default. A comparison with different $\tau$ values could significantly enhance the effectiveness and persuasiveness of the proposed method.
- The proposed method only considers the output probability. What’s the effect of label on the final performance？It is possible that UnGSL decreases the uncertainty of a nodes, but the corresponding predicted label is wrong. For example, message passing from nodes with label $1$ of low uncertainty to another node with label $2$.

**Questions:**

Please refer to the Cons section.

**Reviewer Confidence:**

3: The reviewer is confident but not certain that the evaluation is correct

**Scope:**

4: The work is relevant to the Web and to the track, and is of broad interest to the community

---

### Official Review · Reviewer_oZcn · 2024-11-19

**Novelty:** 4
**Technical Quality:** 4

**Review:**

Summary:
The paper introduces an out-of-the-box Graph Structure Learning (GSL) framework, UnGSL, to address issues in traditional methods, such as neglecting node information quality and the impact of low-quality nodes in symmetric graph structures. The innovation is promising, and the problem background is well-defined. However, the method description is too brief and seems unfinished, while the experimental comparisons and dataset coverage are insufficient. Additionally, there are several formatting issues that affect the overall quality of the paper.

Strengths:
1.	The paper presents a novel idea with the introduction of UnGSL to address the issue of ignoring node quality and the influence of low-quality nodes in symmetric graph structures.
2.	The problem background is clearly presented, and the objectives of the work are well-defined.
3.	Overall, the writing is good and clear.
Weaknesses:
1.	Figure 1: When the GRCN method is first mentioned, a reference should be provided.
2.	Citation formatting: The citation for Shannon Entropy should be written as (Shannon Entropy [20]) rather than (Shannon Entropy [20] ).
3.	Section 2.1 formatting: The last sentence should be corrected to "(e.g., 25%) [6, 12]" instead of "(e.g., 25%).[6, 12]".
4.	Section 3.2 Methodology: The description of the method in Section 3.2 is incomplete and vague. How is the node uncertainty estimated (U)? The authors should provide the formula. How is A∗ used to improve GSL? At least a complete description or example is needed. Additionally, the authors mention that “UnGSL first pretrains the GSL model to estimate node uncertainty,” but it is unclear how pretraining is implemented and how it integrates with downstream tasks.
5.	Lack of Framework Diagram: The paper lacks a clear framework diagram to illustrate how UnGSL works.
6.	Experimental Validation: The authors should validate their method on traditional GNN models to demonstrate the improvement, rather than only focusing on GSL-based models.
7.	Dataset Size: As an efficient method, UnGSL should be validated on larger-scale datasets, as the datasets used in the current experiments only consist of a few thousand nodes at most.
8.	Inconsistent Abbreviations: The use of abbreviations and full terms is inconsistent throughout the paper, e.g., "Fig X" is used initially, but later "Figure 4" appears.
9.	Citation Formatting Issues: The references contain inconsistent formatting, particularly with journal/conference names that should be capitalized.

**Questions:**

1.	How would the results change if other methods of measuring information quality were used?
2.	How exactly is A∗ used to improve GSL in the proposed method?
3.	In Figure 6, why does UnGSL reduce the convergence time of SUBLIME on Cora, but show the opposite effect on Pubmed?

**Reviewer Confidence:**

4: The reviewer is certain that the evaluation is correct and very familiar with the relevant literature

**Scope:**

3: The work is somewhat relevant to the Web and to the track, and is of narrow interest to a sub-community

---

### Official Review · Reviewer_FXu8 · 2024-11-22

**Novelty:** 5
**Technical Quality:** 4

**Review:**

Pros:
1) The motivation of the work is clear.
2) The intuition behind the proposed method is well-illustrated and explained.
3) The structure of the paper is easy-to-follow.
4) The difference from the previous study is highlighted.
5) The experiments and ablation study are quite thorough.

Cons:
1) The pipeline of the use of GSL + GNN for the downstream task is poorly explained. It prevented the full understanding on what is going on in the proposed method.
2) The theoretical part is not persuasive. The theoretical proposition does not support the findings made from it.
3) There are no comments on the connections of the paper with the Web domain (as requires in the Conference guidelines).

**Questions:**

1) In row 16, was it supposed to use the phrase "ambitious information" instead of "ambiguous information"?
2) In formula 2, why does not the loss term "Task" take the optimal graph $\mathbf{S}$ as input? The optimization process of GNN encoder on the downstream task depends significantly on the underlying graph structure from GSL stage. Also, it is unclear what is understood by the regularization loss term? What is usual purpose of regularization procedure in GSL? Moreover, there is a confusion on the train paradigm for GSL+GNN? Do we follow the two-stage scenario and conduct a pre-training of GSL and then train the GNN on the downstream task? Or do we train GSL+GNN in end-to-end fashion?
3) In formula 3, what is $\mathbf{Z}$ matrix that goes to the input of a metric function $\phi(.)$? To transfer the original feature representation of nodes expressed in matrix $\mathbf{X}$ seems to be more logical.
4) In rows 268-270, does $\mathbf{S}_{ij}$ means the directed edge from node $i$ to the node $j$? If yes, are the indices of $\mathbf{S}$ not confused in this text part?
5) In Appendix section A, it is necessary to cite a book/paper that contains a proof for the used log-sum inequality.
6) The statement of Proposition 1 is confusing. The proposition says that the entropy of a node $v_i$ represented as a sum of logits of neighbors is greater than the sum of entropies of neighbors' logits with the coefficients $\eta_j$. However, there is no any estimation of the value range for the coefficients $\eta_j$ in the main paper part. From the proposition, the authors make a conclusion: "If a node is connected to high-uncertainty neighbors, its own uncertainty will inevitably increase after GNN learning". What do you mean under a node uncertainty before GNN layer? If $H(p_i^{\'})$, there are no any inequalities to prove this insight, namely, $H(p_i) - H(p_i^{\'})$ is not estimated directly.
7) When you read the subsection 3.2, it is not clear how the uncertainty of nodes $\mathbf{U}$ is estimated. Do we have the trained GSL model on an unsupervised task and then we refine the learnt graph structure with UnGSL? There is no clearly described pipeline.
8) The row 435 that contains explanation of the meaning of $C_{ij}$ is confusing. Is not the whole $i$-th row of matrix $C$ related to the confidence of the node $v_i$?
9) About "model-agnostic" property. There is no clear explanation in which part of the pipeline the UnGSL can be incorporated.
10) There is a concern on statement "demonstrating its robustness against topological perturbations". As we can see from Figure 5, the performance deteriorates with a very similar behavior as a basic GSL does. The difference is that the initial quality with an addition of UnGSL is greater, thus the further values with different percent of perturbations are greater. However, we can not say that this approach is robust to topological perturbations.

**Reviewer Confidence:**

3: The reviewer is confident but not certain that the evaluation is correct

**Scope:**

4: The work is relevant to the Web and to the track, and is of broad interest to the community

---

### Official Review · Reviewer_ubyQ · 2024-11-29

**Novelty:** 4
**Technical Quality:** 5

**Review:**

# Summary
This paper centers on advancements in Graph Structure Learning (GSL) by addressing fundamental challenges related to node uncertainty and graph asymmetry. Its combination of theoretical insights, practical implementation, and experimental validation makes it a significant contribution to the field of Graph Neural Networks (GNNs).

## Strengths
1. Extensive experiments are conducted, including both homogeneous and heterogeneous graphs.
2. Theoretical analysis is provided.

## Weaknesses
1. The datasets are well-known benchmarks but do not fully represent the complexities of noisy, large-scale, or domain-specific real-world graphs.
2. While the paper claims minimal overhead, learnable thresholds for each node and edge adjustments might increase computational costs for very large or dense graphs.
3. The effectiveness of UnGSL is sensitive to hyperparameters like β, τ, and the initialization of thresholds ϵ. Tuning these parameters for new datasets might require significant effort and expertise.

**Questions:**

## Questions/Comments
1. The learnable node-wise thresholds ϵ play a critical role in distinguishing high- and low-confidence neighbors. Could you elaborate on how these thresholds are initialized and whether their convergence is consistent across datasets?
2. Beyond node quality, have you conducted any experiments or analyses focusing on the quality of labels and edges? In real-world applications, labels may be noisy or incorrect, and edge weights or directions might be imprecise. These factors can significantly impact the performance of GNNs. Can UnGSL be extended to handle such quality issues? If not, do you think designing similar mechanisms to dynamically adjust label or edge quality would be feasible?
3. For heterogeneous graphs, can UnGSL make different adjustments for different types of edges? Given that edges in heterogeneous graphs may represent various relationships or interactions, does UnGSL have the flexibility to account for these distinctions? If not, what modifications would be needed to extend UnGSL to this scenario?
4. There are a few typos or grammatical errors. The authors may need to proofread the manuscript again, including:
   - **Page 2**: In this section, we introduce basic notations and "backgraound" on GNNs and GSL methods.
   - **Page 3**: We then present the proposed "Ucertainty-aware" graph structure...

**Reviewer Confidence:**

3: The reviewer is confident but not certain that the evaluation is correct

**Scope:**

3: The work is somewhat relevant to the Web and to the track, and is of narrow interest to a sub-community

---

### Official Review · Reviewer_s2DJ · 2024-12-03

**Novelty:** 3
**Technical Quality:** 4

**Review:**

This paper is a well-written, incremental contribution to the field of GSL, building on existing work with clear explanations. While the motivation behind the problem is well-articulated, the paper lacks significant novelty or originality, offering mainly incremental improvements. The focus on measuring node quality is valuable and could inspire further research in this direction. However, the experiments are limited to a narrow set of datasets, and the findings may not generalize well to other types of graph structures. Further investigation into the effect of node similarity on performance, particularly by removing dissimilar nodes, could provide deeper insights. Additionally, evaluating the method on a more diverse range of datasets would strengthen the conclusions and help assess the method’s generalizability.

**Questions:**

1- In Figure 1, how does the result change if you eliminate nodes with lower similarity instead of removing random nodes? Intuitively, highly heterophilic nodes are likely to exhibit greater uncertainty or entropy in their class probabilities, as they receive messages from dissimilar nodes. Removing these dissimilar nodes identified by GSL methods could provide stronger support for your hypothesis, allowing us to observe a clearer correlation between dissimilarity and high entropy. If the plot for "eliminating dissimilar nodes" resembles that of "eliminating low-quality neighbors," it could suggest that the marginal gain observed in the latter case is primarily driven by the effect of denoising.

2- Regarding Figure 3: Most nodes in the Cora dataset exhibit a homophily value of 0.8 or higher. As a result, message passing (smoothing) between similar nodes is unlikely to generate significant variations in entropy among neighbors. What observations did you make when plotting the same distribution on a dataset with different homophily characteristics, such as Flickr, which has a more uniform distribution of homophily values?

3- Can you share your results on a more diverse range of datasets in terms of size and homophily?

4- There are some discrepancies in accuracy between the values you reported and those shared by SUBLIME and SLAPS. Could this be due to using different splits or employing hyperparameters that differ from those of the optimal models?

5- Could you highlight the types of nodes that contributed most to the accuracy improvement, such as those with high heterophily or similar characteristics?

**Reviewer Confidence:**

4: The reviewer is certain that the evaluation is correct and very familiar with the relevant literature

**Scope:**

4: The work is relevant to the Web and to the track, and is of broad interest to the community